# Evaluating retinal thickness classification in children: A comparison between pediatric and adult optical coherence tomography databases

**Tian Siew Pua, Mohd Izzuddin Hairol**⊙*

Centre for Community Health Studies (ReaCH), Faculty of Health Sciences, Universiti Kebangsaan Malaysia, Kuala Lumpur, Malaysia

* Izzuddin.hairol@ukm.edu.my

## Abstract

### Purpose

This study investigates the agreement of children's retinal thickness classification by color category between Topcon 3D OCT-1's built-in adult reference data and our new pediatric database and assesses the correlation of retinal thickness with age and spherical equivalent (SE).

### Methods

160 eyes of 160 healthy children (74 boys, 86 girls) aged 6–18 years (mean: 11.60 ± 3.28 years) were evaluated in this cross-sectional study. The peripapillary retinal nerve fibre layer (pRNFL) and macular thickness were determined for the 1st, 5th, 95th, and 99th percentile points. Cohen's κ value and specific agreement between pediatric data and adult reference database were estimated. The correlation between retinal thickness with age and SE was also determined.

### Results

The mean thickness for the total RNFL, average macular, and central macula were 112.05 ±8.65 μm, 280.24±12.46 μm, and 220.55±17.53 μm, respectively. The overall agreement between the classification of the adult database and pediatric data for pRNFL was ≥90%, with discrepancies in 46 out of 150 eyes (30.67%); for macula, it was above 72%, with discrepancies in 93 out of 153 eyes (60.78%); and for ganglion cell complex and ganglion cell + inner plexiform layer (GCIPL) the agreement was above 84% and 85%, respectively. A significant level of agreement between pediatric data and adult reference data was achieved for temporal RNFL (κ = 0.65), macular perifoveal superior (κ = 0.67), and inferior (κ = 0.63) and inferior GCIPL (κ = 0.67). The correlations between age and retinal thickness were not significant (all p>0.05). Most retinal thickness parameters were positively associated with SE (Pearson's coefficient, r = 0.26 to 0.49, all p<0.05).

**Data Availability Statement:** The data file is available at https://figshare.com/articles/dataset/Pua_Hairol_OCT_data_/27014056?file=50781300.

**Funding:** The author(s) received no specific funding for this work.

**Competing interests:** The authors have declared that no competing interests exist.

## Conclusions

The overall agreement for pRNFL and macular thickness measurements in children with the adult reference database was between 72% and 90%. Children's retinal thickness was not significantly correlated with age but was positively associated with spherical equivalent.

## Introduction

Optical coherence tomography (OCT) is a noninvasive imaging technique that uses low-coherence light to capture high-resolution cross-section pictures of ocular tissues. Each retinal layer can be discriminated, and retinal thickness can be measured in vivo and in real time, allowing the retina and the optic nerve to be studied for their morphology and for monitoring disease progression and resolution. Thus, OCT has become the preferred investigative tool in ocular diseases such as glaucoma, retinal diseases, and other optic nerve diseases [1, 2].

OCT imaging technology has experienced rapid evolution and advancement over the decades since its introduction in 1991 [3], from time-domain OCT (TD-OCT) to spectral-domain OCT (SD-OCT) and the latest swept-source OCT (SS-OCT). TD-OCT has a scanning speed of 400 A-scans per second and axial image resolution of 10 μm, SD-OCT has a much higher acquisition rate of 27,000 to 70,000 A-scans per second and axial resolution of 5–7 μm, and SS-OCT has the highest scanning speed of 100,000–400,000 A-scans per second with an axial resolution of 5–7 μm [4, 5]. Since then, OCT image processing time has decreased with the improved quality of the captured images, making it widely used in younger populations in clinical practice.

The OCT reference database derived from the healthy population, such as for the Topcon 3D OCT-1 Maestro [3], enables quantitative comparisons between normal healthy subjects and potentially abnormal observations in diseased or anomalous visual systems. Clinical evaluations of the retinal nerve fibre layers (RNFL) are essential in monitoring and diagnosing optic nerve diseases such as glaucoma. For example, children with autosomal dominant optic atrophy, congenital glaucoma, optic neuritis, or optic nerve hypoplasia [6–9] are reported to have thinner RNFL associated with poorer visual functions. Besides, children with a history of retinopathy of prematurity [10] or exposure to secondhand smoke [11] exhibited thinner RNFL. In addition, the assessment of a patient's macular thickness profiles facilitates the diagnosis and monitoring of macular diseases such as macular edema and macular degeneration. OCT has been used as a diagnostic tool and prognostic indicator in pediatric macular diseases such as foveal hypoplasia [12], albinism [13], infantile nystagmus [14], and amblyopia [15].

In children, there is inconsistency in whether OCT parameters are associated with age. Earlier studies reported either no association between age and RNFL thickness in children [16–18], a positive correlation [19], or a negative correlation [20, 21]. Macular thickness, on the other hand, is typically positively correlated with age [22–24]. Interestingly, myopia and axial length affect RNFL thickness in adults [25] but not in children [21]. Several studies reported ethnic differences in retinal thickness in adults [26, 27].

Thus, accurate evaluation of changes in retinal measurements demands comparing them with reference ranges established for the healthy population. Currently, commercially available OCT devices only include age-normalized databases for individuals older than 18 years, making its application limited to pediatric patients. Consequently, thickness maps for children are generated based on comparisons with the adult database [28]. For Topcon 3D OCT-1 maestro, the established reference database is based on the retinal thickness of 399 healthy adults (age

range 18 to 88 years), where 59% of them were Caucasians [3]. Muñoz-Gallego et al. [29] recently reported that using the Topcon OCT, the likelihood of detecting abnormal macular measurements in children with the adult database was only 49%, highlighting the need for more accurate reference values to improve children's detection of abnormal macular thicknesses in children.

The primary objective of this study was to estimate the agreement between macular and peripapillary RNFL (pRNFL) thickness categories in multiethnic children assigned by the current OCT software (using an adult database) and those assigned based on the pediatric data collected in the present study. Our secondary objectives were to determine the association between children's macular and peripapillary thickness with their age and spherical equivalent. We also aimed to produce a reference database of pRNFL and macular thickness in healthy multiethnic Malaysian children for Topcon 3D OCT-1 Maestro 2 (Topcon Corporation, Tokyo, Japan).

## Materials and methods

The protocol of this cross-sectional study was approved by the institution's Research Ethics Committee (UKM/PPI/111/8/JEP-2021-351) and adhered to the tenets of the Declaration of Helsinki. Parents or guardians of all enrolled children provided their written informed consent after a detailed explanation of the study to the parents/guardians and children. The study was conducted at the Optometry Clinic, Universiti Kebangsaan Malaysia, between April 2023 and March 2024.

Healthy children aged 6–18 years old with a history of full-term birth ($\geq$ 37 weeks gestational age) and normal birth weight ($\geq$ 2500g) were invited to participate in this study. All the included children had best corrected visual acuity (BCVA) equal to or better than 0.2 logarithm of the minimum angle of resolution (logMAR) in each eye with interocular difference of not more than 0.1 logMAR, refractive error in sphere equivalent (SE) less than 6.00D (sphere range between -5.75D to + 4.00D, astigmatism $\leq$ 3.00D), stereopsis equal to or better than 120 seconds of arc, cup-to-disc (C/D) ratio of 0.5 or smaller, interocular C/D ratio asymmetry $\leq$ 0.2, and intraocular pressure (IOP) $\leq$ 21mmHg in each eye. Children with any systemic diseases and ocular abnormalities, including tropia, amblyopia, and any ocular pathologies, were excluded.

All participants underwent visual function assessments. The pupils were dilated with two drops of Cyclopentolate 1% at an interval of five minutes. Cycloplegic autorefraction was assessed with a Topcon RM-8800 Auto refractometer after 30 minutes, and sphere equivalents were reported. Then, both eyes of each participant were scanned with the Topcon 3D OCT-1 Maestro 2. A minimum of nine scans were obtained under three different settings for each eye and then averaged for analysis purposes: three 3D Disc (6×6 mm) scans, three 3D Macular (6×6 mm) scans, and three 3D Wide (12×9 mm) scans. Only an OCT image quality score $\geq$35 was accepted, and data from one eye was randomly selected for analysis. The mean values of each sector of the macula and pRNFL from at least two scans were included for analysis, as scans with excessive eye movements were excluded.

The thickness of the peripapillary retinal nerve fibre layer (pRNFL) was calculated between the inner plexiform layer and the outer edge of the RNFL. It was measured using a 3.4 mm diameter temporal-superior-nasal-inferior-temporal (TSNIT) circle automatically placed at the optic disc center. The average RNFL thickness over the four quadrants was reported. Each measurement of the four quadrants was assigned a color label for different percentile distribution of RNFL thickness measurement based on the embedded adult database: red for thicknesses equal to or below percentile 1 ($\leq$ p1), yellow for thicknesses between percentile 1 and

equal to or below percentile 5 ($>$ p1 to $\leq$ p5), green for thicknesses between percentile 5 and equal to or below percentile 95 ($>$ p5 to $\leq$ p95), white for thicknesses above percentile 95 ($>$ p95).

Macular thickness was determined by measuring the distance between the inner limiting membrane and the retinal pigmented epithelium's inner boundary. The thickness was measured for each of the nine regions defined by the Early Treatment Diabetic Retinopathy Study (ETDRS) grid. An average value of central fovea, superior, nasal, inferior, and temporal inner quadrants (parafoveal); superior, nasal, inferior, and temporal outer quadrants (perifoveal) were reported. Each measurement of the nine macular areas was assigned a color label for different percentile distribution of macular thickness measurement based on the embedded adult database: red for thicknesses equal to or below percentile 1 ($\leq$ p1), yellow for thicknesses between percentile 1 and equal to or below percentile 5 ($>$ p1 to $\leq$ p5), green for thicknesses between percentile 5 and equal to or below percentile 95 ($>$ p5 to $\leq$ p95), orange for thicknesses between percentile 95 and equal to or below percentile 99 ($>$ p95 to $\leq$ p99), and pink for thicknesses above percentile 99 ($>$ p99).

Macula ganglion cell and inner plexiform layer thickness (GCIPL) and macula ganglion cell complex (GCC = GCIPL+RNFL) were obtained from the 3D Wide protocol. GCC and GCIPL were measured across the macula's six-sector circle. An average value of the entire sector (superior, superior nasal, superior temporal, inferior, inferior nasal, inferior temporal) was reported. Each measurement of the six sectors was assigned a color label for different percentile distribution of GCIPL and GCC thickness measurement based on the embedded adult database: red for thicknesses equal to or below percentile 1 ($<$ p1), yellow for thicknesses between percentile 1 and equal to or below percentile 5 ($>$ p1 to $\leq$ p5), and green for thicknesses above percentile 5 ($>$ p5).

## Data analysis

Statistical analyses were conducted using the Statistical Package for the Social Sciences (version 25). The normality of the distribution of the study sample was assessed either by the Kolmogorov-Smirnov test, skewness or kurtosis statistics, or by visual inspection of the histogram. Data for all measurements were presented as mean and standard deviation. Pearson correlation analysis was used to determine the relationship between macular and pRNFL thickness with age and SE. P values less than 0.05 were considered to be statistically significant.

To estimate the agreement of the macular and pRNFL thickness between the children's data and the current adult database, all retinal thickness parameter values were determined for the 1st, 5th, 95th, and 99th percentile points. The proportion of agreement [29–31] was calculated based on the classification of retinal thickness according to percentiles from the adult database and pediatric data from this study. The agreement was calculated in the form of a percentage based on the proportion of total study subjects classified by Topcon 3D OCT-1 as having the same percentile classification from the actual distribution of study sample data. Observed agreement (OA) refers to overall agreement. Specific agreement (SA) was calculated to determine the measurement agreement less than or equal to the 5th percentile (SAp$\leq$5) and more than the 95th percentile (SAp$>$95). Cohen's Kappa ($\kappa$) analysis was also used to determine the agreement of macular and peripapillary RNFL thickness classified as green (thickness $>$ p5 to $\leq$ p95) according to percentiles between the adult database classified by the instrument and the pediatric data from this study. $\kappa$ value ranges between -1 and 1 where $\kappa$ = 1 indicates perfect agreement and $\kappa \geq 0.6$ indicates a significant level of agreement [32].

## Results

The mean age of the 160 participants included was 11.60 ± 3.28 years (range 6.03–18.37 years). Among them, 46.25% (74) were boys, while 53.75% (86) were girls. The participants were comprised of 72.5% (116) Malay, 21.88% (35) Chinese, and 5.62% (9) Indian or other ethnicity children. This approximates the ethnic distribution of Malaysia's resident population [33]. The mean cycloplegic SE, BCVA, stereopsis, and IOP of the participants were -1.31 ± 1.90D, 0.01± 0.06 logMAR, 58.37± 22.80 seconds of arc, and 17.69 ± 1.98 mmHg, respectively.

### Retinal nerve fibre layer thickness measurements

A total of 150 eyes were included in the analysis for pRNFL, and ten subjects were excluded due to excessive eye movement for the 3D disc scan. Table 1 shows the mean for total RNFL thickness and various sectors, including the 1st, 5th, 95th, and 99th percentiles. The thickest sector was the superior quadrant (145.11±14.51 μm), and the thinnest sector was the nasal quadrant (79.13±12.78 μm). The differences between the sectorial RNFL thicknesses were significant [$F_{(3, 596)}$ = 1169.21, p<0.001]. Post hoc analyses with Games-Howell revealed that superior RNFL was significantly thicker than nasal RNFL (p<0.001) and temporal RNFL (p<0.001); inferior RNFL was also significantly thicker than nasal RNFL (p<0.001) and temporal RNFL (p < 0.001). There was no significant difference between superior RNFL and inferior RNFL and between nasal RNFL and temporal RNFL.

The correlation between age and total RNFL thickness was not significant (r = -0.02, p = 0.774). However, weak but positive correlations were found between SE and total (r = 0.26, p = 0.002), nasal (r = 0.32, p<0.001), and inferior (r = 0.36, p<0.001) RNFL thicknesses, and a weak but negative correlation between SE and temporal (r = -0.31, p < 0.001) RNFL thickness. Correlations between SE groups and RNFL thickness are shown in Table 2, where a significant correlation was found only in the myopic SE group.

**Agreement between the reference adult database and the pediatric database.** For peripapillary RNFL thickness measurements, the overall agreement (OA) between the classification of the adult database and pediatric data was 90% and above. Fig 1 shows the detailed breakdown of the classification of adult and pediatric databases into four different percentile-based categories (≤ p1, > p1 to ≤ p5, > p5 to ≤ p95, and > p95). Discrepancies were found in 46 out of 150 eyes (30.67%) with 56 measurements. Forty-six out of the 56 measurements (82.14%) were assigned to the lower percentile category, while the remaining 10 measurements (17.86%) were assigned to the higher percentile category with the pediatric database. A total of 54 measurements were considered abnormal (≤ p5 and > p95) based on the pediatric database, and 34 measurements (62.96%) were classified as the same color category with the reference adult database.

For the superior, nasal, inferior, and temporal RNFL, the two databases agreed regarding the correct percentile classification for 136, 135, 137, and 140 out of 150 eyes, respectively.

**Table 1. Peripapillary RNFL thickness profile.**

| RNFL (μm) | Mean ± SD | Median | 1st | 5th | 95th | 99th | Range |
|---|---|---|---|---|---|---|---|
| Total | 112.05 ± 8.65 | 111.83 | 87.28 | 98.00 | 125.82 | 138.73 | 81–144 |
| Superior | 145.11 ± 14.51 | 145.00 | 105.51 | 117.00 | 169.00 | 174.47 | 105–176 |
| Nasal | 79.13 ± 12.78 | 78.00 | 49.00 | 60.55 | 102.45 | 117.94 | 49–121 |
| Inferior | 144.89 ± 15.61 | 142.00 | 113.51 | 123.00 | 174.00 | 196.78 | 113–208 |
| Temporal | 79.32 ± 11.06 | 79.00 | 55.53 | 61.55 | 99.80 | 111.47 | 54–113 |

**Table 2. Correlation between RNFL thickness with SE groups.**

| RNFL | Myope (n = 91) | | | Emmetrope (n = 38) | | | Hyperope (n = 21) | | |
|---|---|---|---|---|---|---|---|---|---|
| | Mean SE: -2.45 ± 1.38 | | | Mean SE: +0.16 ± 0.27 | | | Mean SE: +1.30 ± 0.64 | | |
| | Mean ± SD | $r^a$ | p-value | Mean ± SD | $r^a$ | p-value | Mean ± SD | $r^a$ | p-value |
| Total | 110.91 ± 7.98 | 0.21 | 0.043* | 112.38 ± 8.57 | 0.09 | 0.607 | 116.40 ± 10.41 | 0.17 | 0.467 |
| Superior | 144.21 ± 14.67 | 0.15 | 0.152 | 144.45 ± 13.66 | 0.24 | 0.149 | 150.19 ± 14.96 | -0.06 | 0.797 |
| Nasal | 76.31 ± 11.86 | 0.22 | 0.038* | 82.29 ± 12.63 | -0.14 | 0.389 | 85.67 ± 13.73 | 0.15 | 0.521 |
| Inferior | 141.95 ± 14.63 | 0.32 | 0.002^ | 147.84 ± 14.96 | 0.17 | 0.299 | 152.29 ± 18.02 | 0.37 | 0.101 |
| Temporal | 81.55 ± 11.33 | -0.26 | 0.014* | 75.00 ± 9.91 | -0.10 | 0.537 | 77.48 ± 9.53 | -0.08 | 0.737 |

$^a r$, Pearson coefficient

*, statistically significant, $p < 0.05$ (two-tailed)

^, statistically significant, $p < 0.01$ (two-tailed)

Only the temporal RNFL (κ = 0.65) sector achieved a significant level of agreement (κ ≥ 0.6). For the other sectors, the κ values were 0.54 (superior), 0.49 (nasal), and 0.51 (inferior).

## Macular thickness measurements

A total of 153 participants with 153 eyes were included in the macular analysis, and seven subjects were excluded due to excessive eye movement during the 3D macular scan. The average macular thickness was 280.24±12.46 μm, the minimum foveal thickness was 176.16±14.22 μm, and the total macular volume was 7.92±0.35 mm³. Table 3 shows the distribution of macular thickness according to the ETDRS grid.

Age had no significant correlation with all macular parameters. There was a moderate and significant positive correlation between SE and average macular thickness (r = 0.43, p < 0.001), total volume (r = 0.43, p < 0.001), thickness of the outer superior (r = 0.42, p < 0.001), outer inferior (r = 0.45, p < 0.001), and outer temporal (r = 0.49, p < 0.001) sectors. At the same time, there was a weak but significant positive correlation between SE and thickness of the inner superior (r = 0.29, p < 0.001), inner nasal (r = 0.28, p < 0.001), inner inferior (r = 0.28, p = 0.001), inner temporal (r = 0.26, p = 0.001), and outer nasal (r = 0.34, p < 0.001) sectors. No significant correlation was found between SE and minimum foveal thickness (r = -0.03, p = 0.705) and centre ETDRS thickness (r = 0.02, p = 0.781). Table 4 summarizes the correlation between macular parameters with age and SE.

Statistically significant positive correlations were found in myopic SE for all the macular parameters except minimum foveal and centre ETDRS thickness. No significant correlation was found between the emmetrope and hyperope groups, as shown in Table 5.

**Agreement between the reference adult database and the pediatric database.** The OA between the classification of adult and pediatric databases was above 72% for all macular ETDRS measurements. The detailed breakdown of the classification of the adult and pediatric database into five different percentile-based categories is shown in Fig 2. Discrepancies were found in 93 out of 153 eyes (60.78%) with 289 measurements, where 281 out of 289 (97.23%) measurements were assigned to the higher percentile category, while the remaining eight measurements (2.77%) were assigned to the lower percentile category with pediatric database. A total of 134 measurements were considered abnormal (≤ p5 and > p95) based on the pediatric data, and only 41 measurements (30.60%) were classified as the same color category with the reference adult database.

From a total of 153 eyes, the two databases were in agreement regarding the correct percentile classification for 111 eyes in the centre, 119 eyes in the inner superior sector, 115 eyes in

**Superior RNFL**

| Reference value Pediatric | | Reference value Adult | |
|---|---|---|---|
| Percentile | n | n | Percentile |
| ≤p1 | | 0 | ≤p1 |
| >p1 to ≤p5 | | | |
| >p5 to ≤p95 | | | |
| >p95 | | | |
| ≤p1 | 1 | 4 | >p1 to ≤p5 |
| >p1 to ≤p5 | 3 | | |
| >p5 to ≤p95 | | | |
| >p95 | | | |
| ≤p1 | | 135 | >p5 to ≤p95 |
| >p1 to ≤p5 | 4 | | |
| >p5 to ≤p9 | 131 | | |
| >p95 | | | |
| ≤p1 | | 20 | >p95 |
| >p1 to ≤p5 | | | |
| >p5 to ≤p9 | 14 | | |
| >p95 | 6 | | |

Specific agreement (%) between
Peadiatric and adult for ≤p5 = 66.67%
Pediatric and adult for >p95 = 46.15%

**Inferior RNFL**

| Reference value Pediatric | | Reference value Adult | |
|---|---|---|---|
| Percentile | n | n | Percentile |
| ≤p1 | | 0 | ≤p1 |
| >p1 to ≤p5 | | | |
| >p5 to ≤p95 | | | |
| >p95 | | | |
| ≤p1 | 1 | 5 | >p1 to ≤p5 |
| >p1 to ≤p5 | 4 | | |
| >p5 to ≤p95 | | | |
| >p95 | | | |
| ≤p1 | | 139 | >p5 to ≤p95 |
| >p1 to ≤p5 | 2 | | |
| >p5 to ≤p9 | 137 | | |
| >p95 | | | |
| ≤p1 | | 15 | >p95 |
| >p1 to ≤p5 | | | |
| >p5 to ≤p9 | 7 | | |
| >p95 | 8 | | |

Specific agreement (%) between
Peadiatric and adult for ≤p5 = 83.33%
Pediatric and adult for >p95 = 69.57%

**Nasal RNFL**

| Reference value Pediatric | | Reference value Adult | |
|---|---|---|---|
| Percentile | n | n | Percentile |
| ≤p1 | | 0 | ≤p1 |
| >p1 to ≤p5 | | | |
| >p5 to ≤p95 | | | |
| >p95 | | | |
| ≤p1 | 2 | 2 | >p1 to ≤p5 |
| >p1 to ≤p5 | | | |
| >p5 to ≤p95 | | | |
| >p95 | | | |
| ≤p1 | | 140 | >p5 to ≤p95 |
| >p1 to ≤p5 | 6 | | |
| >p5 to ≤p9 | 134 | | |
| >p95 | | | |
| ≤p1 | | 17 | >p95 |
| >p1 to ≤p5 | | | |
| >p5 to ≤p9 | 9 | | |
| >p95 | 8 | | |

Specific agreement (%) between
Peadiatric and adult for ≤p5 = 40%
Pediatric and adult for >p95 = 64%

**Temporal RNFL**

| Reference value Pediatric | | Reference value Adult | |
|---|---|---|---|
| Percentile | n | n | Percentile |
| ≤p1 | | 0 | ≤p1 |
| >p1 to ≤p5 | | | |
| >p5 to ≤p95 | | | |
| >p95 | | | |
| ≤p1 | 1 | 15 | >p1 to ≤p5 |
| >p1 to ≤p5 | 8 | | |
| >p5 to ≤p9 | 6 | | |
| >p95 | | | |
| ≤p1 | | 139 | >p5 to ≤p95 |
| >p1 to ≤p5 | | | |
| >p5 to ≤p9 | 137 | | |
| >p95 | 2 | | |
| ≤p1 | | 5 | >p95 |
| >p1 to ≤p5 | | | |
| >p5 to ≤p95 | | | |
| >p95 | 5 | | |

Specific agreement (%) between
Peadiatric and adult for ≤p5 = 75%
Pediatric and adult for >p95 = 83.33%

**Fig 1. Agreement between adult and pediatric OCT reference value classification.** Specific agreement between adult and pediatric percentiles equal to or below percentile 5 and above percentile 95 for pRNFL in healthy children aged 6–18 years (n = 150).

the inner nasal sector, 119 eyes in the inner inferior sector, 126 eyes in the inner temporal sector, 141 eyes in the outer superior sector, 135 eyes in the outer nasal sector, 142 eyes in the outer inferior sector, and 132 eyes in the outer temporal sector. Only outer superior and outer

Table 3. Macular thickness profile based on ETDRS quadrants.

| ETDRS Quadrants (µm) | Mean ± SD | Mean | 1st | 5th | 95th | 99th | Range |
|---|---|---|---|---|---|---|---|
| Centre | 220.55 ± 17.53 | 222.00 | 176.16 | 192.40 | 251.60 | 269.74 | 174–280 |
| Inner Superior | 308.80 ± 13.37 | 308.00 | 280.54 | 285.70 | 333.30 | 349.38 | 280–351 |
| Inner Nasal | 308.85 ± 14.26 | 308.00 | 273.00 | 287.00 | 333.00 | 350.98 | 273–358 |
| Inner Inferior | 305.90 ± 13.17 | 305.00 | 273.16 | 284.70 | 327.30 | 343.84 | 271–346 |
| Inner Temporal | 296.43 ± 12.66 | 296.00 | 269.70 | 276.00 | 319.00 | 333.30 | 267–336 |
| Outer Superior | 278.29 ± 13.96 | 277.00 | 245.00 | 257.00 | 304.60 | 312.38 | 245–314 |
| Outer Nasal | 294.28 ± 15.25 | 295.00 | 254.78 | 269.40 | 320.00 | 329.22 | 251–333 |
| Outer Inferior | 268.90 ± 13.79 | 270.00 | 232.54 | 247.40 | 291.00 | 301.46 | 232–302 |
| Outer Temporal | 259.37 ± 13.63 | 259.00 | 227.00 | 237.80 | 284.30 | 293.22 | 227–297 |

inferior sectors achieved significant levels of agreement between adult and pediatric database classification (κ values 0.67 and 0.63, respectively). The κ values for the centre, inner superior, inner nasal, inner inferior, inner temporal, outer nasal, and outer temporal were 0.16, 0.24, 0.23, 0.21, 0.33, 0.49, and 0.44, respectively.

## GCC and GCIPL thicknesses measurements

Measurements from 157 eyes were included in the analysis for GCC and GCIPL thickness, while three subjects were excluded due to excessive eye movement for the 3D Wide scan. The distribution of the GCC and GCIPL thicknesses is summarized in Table 6.

Pearson's correlation analysis showed that age only had a weak but significant positive correlation with the superior nasal GCC thickness (r = 0.18, p = 0.026) and inferior nasal GCC thickness (r = 0.19, p = 0.016). There was no significant correlation between age and non-nasal GCC thicknesses. SE only showed a significant but weak positive correlation with superior temporal GCC thickness (r = 0.36, p < 0.001) and inferior temporal GCC thickness (r = 0.33, p < 0.001) but no correlation with non-temporal GCC sectors. For GCIPL thickness, no

Table 4. Correlation between macular parameters with age and SE.

| Macular Parameters | Age | | SE | |
|---|---|---|---|---|
| | $r^a$ | p-value | $r^a$ | p-value |
| Average thickness | -0.04 | 0.654 | 0.43 | < 0.001** |
| Minimum foveal | 0.09 | 0.247 | -0.03 | 0.705 |
| Total volume | -0.04 | 0.654 | 0.43 | < 0.001** |
| Centre | 0.15 | 0.064 | 0.02 | 0.781 |
| Inner superior | 0.06 | 0.458 | 0.29 | < 0.001** |
| Inner nasal | 0.10 | 0.206 | 0.28 | < 0.001** |
| Inner inferior | 0.11 | 0.183 | 0.28 | 0.001^ |
| Inner temporal | 0.06 | 0.441 | 0.26 | 0.001^ |
| Outer superior | -0.11 | 0.166 | 0.42 | < 0.001** |
| Outer nasal | 0.03 | 0.729 | 0.34 | < 0.001** |
| Outer inferior | -0.14 | 0.086 | 0.45 | < 0.001** |
| Outer temporal | -0.09 | 0.251 | 0.49 | < 0.001** |

[a]r, Pearson coefficient

^, statistically significant p < 0.01 (two-tailed)

**, statistically significant p < 0.001 (two-tailed)

**Table 5. Correlation between macular parameters with SE groups.**

| Macula parameters | Myope (n = 97) | | | Emmetrope (n = 37) | | | Hyperope (n = 19) | | |
|---|---|---|---|---|---|---|---|---|---|
| | Mean SE: -2.49 ± 1.41 | | | Mean SE: +0.17 ± 0.28 | | | Mean SE: +1.34 ± 0.66 | | |
| | Mean ± SD | $r^a$ | p-value | Mean ± SD | $r^a$ | p-value | Mean ± SD | $r^a$ | p-value |
| Average thickness | 277.44 ± 11.19 | 0.36 | < 0.001** | 282.23 ± 12.68 | 0.24 | 0.146 | 290.67 ± 12.56 | 0.06 | 0.817 |
| Minimum foveal | 177.41 ± 15.25 | 0.12 | 0.231 | 174.65 ± 11.65 | 0.21 | 0.203 | 172.68 ± 13.03 | -0.03 | 0.892 |
| Total volume | 7.84 ± 0.32 | 0.36 | < 0.001** | 7.98 ± 0.36 | 0.24 | 0.146 | 8.22 ± 0.36 | 0.06 | 0.813 |
| Centre | 221.42 ± 18.51 | 0.15 | 0.147 | 219.19 ± 15.09 | 0.20 | 0.228 | 218.74 ± 17.29 | -0.11 | 0.647 |
| Inner superior | 306.76 ± 12.44 | 0.24 | 0.020* | 310.89 ± 13.30 | 0.26 | 0.120 | 315.16 ± 16.02 | 0.09 | 0.711 |
| Inner nasal | 306.84 ± 14.06 | 0.24 | 0.020* | 310.43 ± 13.89 | 0.24 | 0.161 | 316.05 ± 13.99 | 0.03 | 0.914 |
| Inner inferior | 304.18 ± 12.51 | 0.24 | 0.019* | 306.86 ± 13.43 | 0.20 | 0.235 | 312.84 ± 14.21 | 0.12 | 0.639 |
| Inner temporal | 294.84 ± 11.94 | 0.25 | 0.014* | 297.95 ± 13.09 | 0.19 | 0.250 | 301.63 ± 14.24 | 0.07 | 0.793 |
| Outer superior | 274.99 ± 12.52 | 0.32 | 0.001^ | 280.30 ± 14.07 | 0.22 | 0.202 | 291.26 ± 13.05 | -0.08 | 0.745 |
| Outer nasal | 291.62 ± 13.77 | 0.29 | 0.004^ | 295.76 ± 17.14 | 0.21 | 0.213 | 305.00 ± 14.18 | 0.09 | 0.705 |
| Outer inferior | 265.68 ± 12.67 | 0.37 | < 0.001** | 271.16 ± 13.27 | 0.29 | 0.082 | 280.89 ± 13.39 | 0.03 | 0.905 |
| Outer temporal | 255.63 ± 11.98 | 0.41 | < 0.001** | 262.46 ± 12.60 | 0.17 | 0.321 | 272.42 ± 14.54 | 0.07 | 0.775 |

$^a r$, Pearson coefficient

*, statistically significant, p < 0.05 (two-tailed)

^, statistically significant, p < 0.01 (two-tailed)

**, statistically significant, p < 0.001 (two-tailed)

correlation with age was found for all six sectors. However, SE was significantly positively correlated with all six sectors of GCIPL thickness (superior temporal: r = 0.35, p < 0.001; superior: r = 0.37, p < 0.001; superior nasal: r = 0.41, p < 0.001; inferior nasal: r = 0.42, p < 0.001; inferior: r = 0.43, p < 0.001; and inferior temporal: r = 0.35, p < 0.001). A significant weak positive correlation was found between inferior and temporal GCC and myopic SE; but a significant moderate negative correlation was found between all sectors of GCC and hyperopic SE. For GCIPL, significant weak to moderate positive correlations were found with myopic SE for all sectors. The summary of the correlations between GCC and GCIPL with refraction groups is shown in Table 7.

**Agreement between the reference adult database and the pediatric database.** For all GCC and GCIPL measurements, the OA between the classification of the adult database and the pediatric data was above 84% and 85%, respectively. The detailed breakdown of the classification of adult and pediatric databases into three different percentile-based categories (≤p1, >p1 to ≤p5, and >p5) are shown in Figs 3 and 4 for GCC and GCPIL measurements, respectively.

For GCC thickness, discrepancies were found with 120 measurements in 49 out of 157 eyes (31.21%). All 120 measurements were assigned to higher percentile categories with the pediatric database. A total of 45 measurements were considered abnormal (≤p5) based on the pediatric database, and only 21 measurements (46.67%) were classified as the same color category with the reference adult database. For GCIPL thickness, discrepancies were found in 113 measurements in 43 out of 157 eyes (27.39%). All 113 measurements were assigned to higher percentile categories with the pediatric database. A total of 36 measurements were considered abnormal (≤p5) based on the pediatric database, and only 14 measurements (38.89%) were classified as the same color category with the reference adult database.

For GCC thickness, from a total of 157 eyes, the two databases agreed regarding the correct percentile classification for 138 eyes in the superior temporal sector, 142 eyes in the superior sector, 133 eyes in the superior nasal sector, 144 eyes in the inferior nasal sector, 149 eyes in

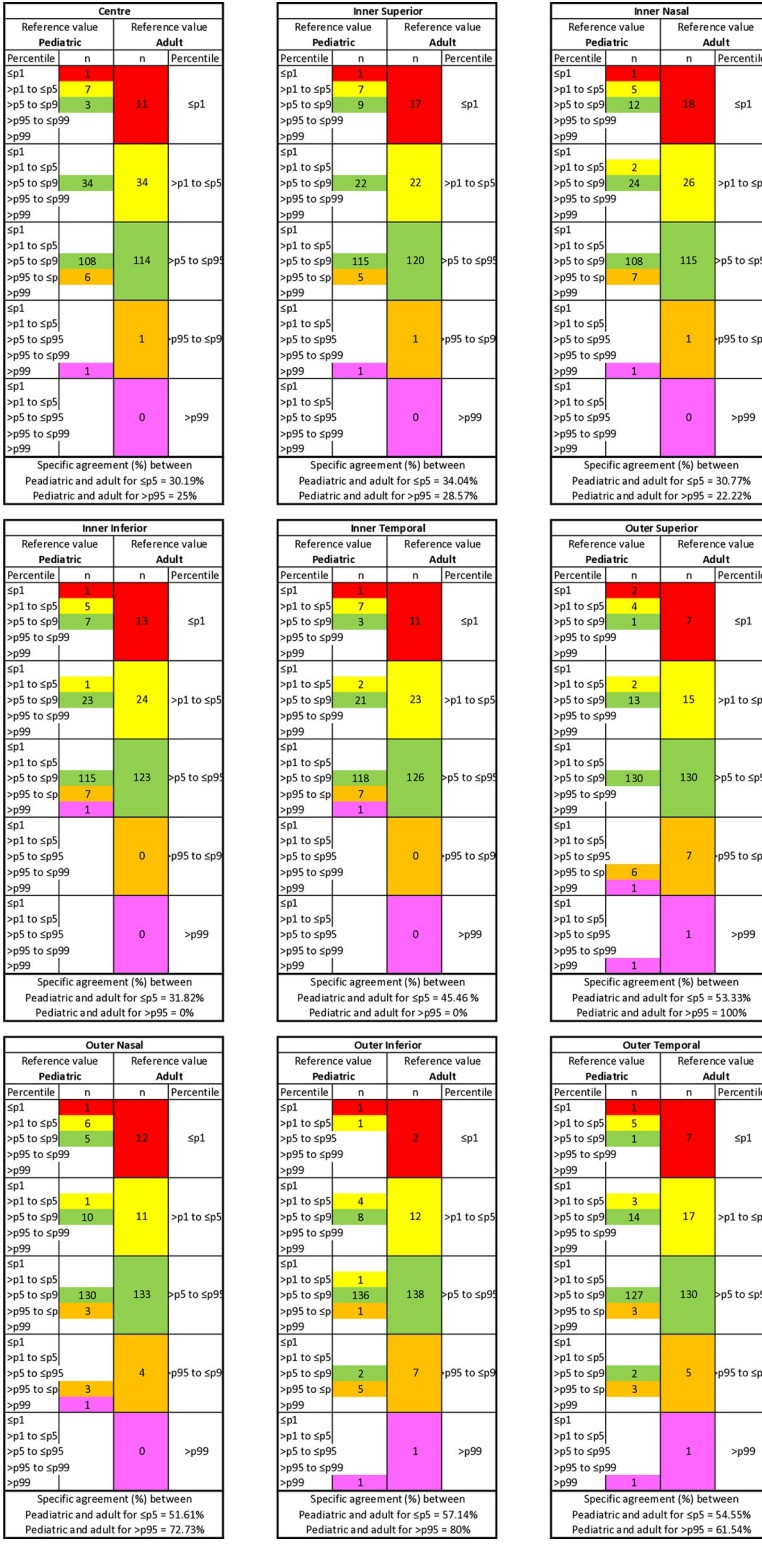

**Fig 2. Agreement between adult and pediatric OCT reference value classification.** Specific agreement between adult and pediatric percentiles equal to or below percentile 5 and above percentile 95 for macular ETDRS thickness in healthy children aged 6–18 years (n = 153).

**Table 6. GCC and GCIPL thickness profiles.**

| | Quadrants | Mean ± SD | Median | 1st | 5th | 95th | 99th | Range |
|---|---|---|---|---|---|---|---|---|
| GCC | Superior Temporal | 96.13 ± 6.26 | 96.00 | 78.16 | 85.90 | 105.20 | 117.10 | 77–120 |
| | Superior | 109.46 ± 6.96 | 109.00 | 89.32 | 98.90 | 123.00 | 129.94 | 87–134 |
| | Superior Nasal | 120.17 ± 7.43 | 120.00 | 95.16 | 107.90 | 133.00 | 141.52 | 94–145 |
| | Inferior Nasal | 120.85 ± 7.98 | 121.00 | 92.00 | 108.00 | 135.00 | 141.52 | 92–145 |
| | Inferior | 107.91 ± 6.92 | 108.00 | 89.48 | 98.00 | 118.00 | 132.78 | 86–138 |
| | Inferior Temporal | 98.78 ± 5.98 | 99.00 | 82.32 | 90.00 | 107.00 | 118.10 | 80–121 |
| GCIPL | Superior Temporal | 72.77 ± 5.57 | 73.00 | 56.16 | 63.00 | 80.20 | 91.10 | 55–94 |
| | Superior | 72.77 ± 5.02 | 73.00 | 57.58 | 65.00 | 81.00 | 88.52 | 57–92 |
| | Superior Nasal | 77.83 ± 5.71 | 79.00 | 57.58 | 69.90 | 88.00 | 92.68 | 57–95 |
| | Inferior Nasal | 76.47 ± 5.36 | 76.00 | 57.74 | 68.00 | 86.00 | 90.52 | 56–94 |
| | Inferior | 69.51 ± 4.70 | 69.00 | 56.48 | 62.90 | 78.00 | 82.52 | 53–86 |
| | Inferior Temporal | 74.30 ± 5.42 | 74.00 | 59.32 | 65.00 | 82.00 | 91.10 | 57–94 |

the inferior sector, and 140 eyes in the inferior temporal sector. The κ values for superior temporal, superior, superior nasal, inferior nasal, inferior, and inferior temporal were 0.38, 0.48, 0.32, 0.52, 0.58, and 0.47, respectively. No sector of GCC achieved a significant level of agreement between adult and pediatric database classification. For GCIPL thickness, from a total of 157 eyes, the two databases agreed regarding the correct percentile classification for 141 eyes in the superior temporal sector, 140 eyes in the superior sector, 135 eyes in the superior nasal sector, 146 eyes in the inferior nasal sector, 150 eyes in the inferior sector, and 139 eyes in the inferior temporal sector. The κ values for superior temporal, superior, superior nasal, inferior nasal, inferior, and inferior temporal were 0.39, 0.38, 0.34, 0.45, 0.67, and 0.28, respectively.

**Table 7. Correlation between GCC and GCIPL with SE groups.**

| Quadrants | Myope (n = 97) | | | Emmetrope (n = 40) | | | Hyperope (n = 20) | | |
|---|---|---|---|---|---|---|---|---|---|
| | Mean SE: -2.51 ± 1.40 | | | Mean SE: +0.17 ± 0.27 | | | Mean SE: +1.24 ± 0.58 | | |
| | Mean ± SD | $r^a$ | Pp-value | Mean ± SD | $r^a$ | p-value | Mean ± SD | $r^a$ | p-value |
| **GCC** | | | | | | | | | |
| Superior temporal | 94.88 ± 6.10 | 0.32 | 0.001^ | 97.03 ± 5.40 | 0.23 | 0.156 | 100.40 ± 6.68 | -0.46 | 0.043* |
| Superior | 108.95 ± 7.13 | 0.14 | 0.174 | 109.23 ± 6.05 | 0.21 | 0.196 | 112.40 ± 7.45 | -0.57 | 0.008^ |
| Superior nasal | 120.31 ± 7.77 | 0.11 | 0.297 | 119.13 ± 6.39 | 0.24 | 0.144 | 121.55 ± 7.77 | -0.67 | 0.001^ |
| Inferior nasal | 121.07 ± 8.44 | 0.07 | 0.505 | 118.80 ± 6.35 | 0.30 | 0.059 | 123.85 ± 7.84 | -0.67 | 0.001^ |
| Inferior | 107.72 ± 7.14 | 0.21 | 0.040* | 106.75 ± 6.05 | 0.30 | 0.062 | 111.15 ± 6.81 | -0.57 | 0.009^ |
| Inferior temporal | 97.82 ± 5.93 | 0.31 | 0.002^ | 98.75 ± 4.56 | 0.20 | 0.212 | 103.50 ± 6.74 | -0.47 | 0.038* |
| **GCIPL** | | | | | | | | | |
| Superior temporal | 71.69 ± 5.53 | 0.31 | 0.002^ | 73.45 ± 4.59 | 0.22 | 0.165 | 76.65 ± 5.89 | -0.42 | 0.063 |
| Superior | 71.64 ± 4.89 | 0.30 | 0.003^ | 73.70 ± 4.36 | 0.23 | 0.155 | 76.40 ± 5.01 | -0.45 | 0.047* |
| Superior nasal | 76.57 ± 5.72 | 0.39 | < 0.001** | 79.18 ± 4.95 | 0.32 | 0.046* | 81.25 ± 5.28 | -0.41 | 0.072 |
| Inferior nasal | 75.28 ± 5.37 | 0.40 | < 0.001** | 77.80 ± 4.43 | 0.29 | 0.068 | 79.60 ± 5.36 | -0.41 | 0.076 |
| Inferior | 68.38 ± 4.50 | 0.37 | < 0.001** | 70.25 ± 4.09 | 0.24 | 0.144 | 73.50 ± 4.55 | -0.37 | 0.112 |
| Inferior temporal | 73.37 ± 5.36 | 0.34 | 0.001^ | 74.33 ± 4.11 | 0.19 | 0.239 | 78.75 ± 6.05 | -0.41 | 0.070 |

[a]$r$, Pearson coefficient

*, statistically significant, $p < 0.05$ (two-tailed)

^, statistically significant, $p < 0.01$ (two-tailed)

**, statistically significant, $p < 0.001$ (two-tailed)

| Superior Temporal | | | |
|---|---|---|---|
| Reference value **Pediatric** | | Reference value **Adult** | |
| Percentile | n | n | Percentile |
| ≤p1 | 1 | 13 | ≤p1 |
| >p1 to ≤p5 | 8 | | |
| >p5 | 4 | | |
| ≤p1 | | 17 | >p1 to ≤p5 |
| >p1 to ≤p5 | | | |
| >p5 | 17 | | |
| ≤p1 | | 128 | >p5 to ≤p95 |
| >p1 to ≤p5 | | | |
| >p5 | 128 | | |
| Specific agreement (%) between Peadiatric and adult for ≤p5 = 46.15% | | | |

| Superior | | | |
|---|---|---|---|
| Reference value **Pediatric** | | Reference value **Adult** | |
| Percentile | n | n | Percentile |
| ≤p1 | 1 | 6 | ≤p1 |
| >p1 to ≤p5 | 5 | | |
| >p5 | | | |
| ≤p1 | | 19 | >p1 to ≤p5 |
| >p1 to ≤p5 | 2 | | |
| >p5 | 17 | | |
| ≤p1 | | 133 | >p5 to ≤p95 |
| >p1 to ≤p5 | | | |
| >p5 | 133 | | |
| Specific agreement (%) between Peadiatric and adult for ≤p5 = 48.49% | | | |

| Superior Nasal | | | |
|---|---|---|---|
| Reference value **Pediatric** | | Reference value **Adult** | |
| Percentile | n | n | Percentile |
| ≤p1 | 1 | 13 | ≤p1 |
| >p1 to ≤p5 | 7 | | |
| >p5 | 5 | | |
| ≤p1 | | 20 | >p1 to ≤p5 |
| >p1 to ≤p5 | | | |
| >p5 | 20 | | |
| ≤p1 | | 125 | >p5 to ≤p95 |
| >p1 to ≤p5 | | | |
| >p5 | 125 | | |
| Specific agreement (%) between Peadiatric and adult for ≤p5 = 39.02% | | | |

| Inferior Nasal | | | |
|---|---|---|---|
| Reference value **Pediatric** | | Reference value **Adult** | |
| Percentile | n | n | Percentile |
| ≤p1 | 1 | 4 | ≤p1 |
| >p1 to ≤p5 | 3 | | |
| >p5 | | | |
| ≤p1 | | 19 | >p1 to ≤p5 |
| >p1 to ≤p5 | 4 | | |
| >p5 | 15 | | |
| ≤p1 | | 135 | >p5 to ≤p95 |
| >p1 to ≤p5 | | | |
| >p5 | 135 | | |
| Specific agreement (%) between Peadiatric and adult for ≤p5 = 51.61% | | | |

| Inferior | | | |
|---|---|---|---|
| Reference value **Pediatric** | | Reference value **Adult** | |
| Percentile | n | n | Percentile |
| ≤p1 | 1 | 3 | ≤p1 |
| >p1 to ≤p5 | 2 | | |
| >p5 | | | |
| ≤p1 | | 11 | >p1 to ≤p5 |
| >p1 to ≤p5 | 3 | | |
| >p5 | 8 | | |
| ≤p1 | | 144 | >p5 to ≤p95 |
| >p1 to ≤p5 | | | |
| >p5 | 144 | | |
| Specific agreement (%) between Peadiatric and adult for ≤p5 = 60% | | | |

| Inferior Temporal | | | |
|---|---|---|---|
| Reference value **Pediatric** | | Reference value **Adult** | |
| Percentile | n | n | Percentile |
| ≤p1 | 1 | 6 | ≤p1 |
| >p1 to ≤p5 | 5 | | |
| >p5 | | | |
| ≤p1 | | 22 | >p1 to ≤p5 |
| >p1 to ≤p5 | 4 | | |
| >p5 | 18 | | |
| ≤p1 | | 130 | >p5 to ≤p95 |
| >p1 to ≤p5 | | | |
| >p5 | 130 | | |
| Specific agreement (%) between Peadiatric and adult for ≤p5 = 52.63% | | | |

**Fig 3. Agreement between adult and pediatric OCT reference value classification.** Specific agreement between adult and pediatric percentiles equal to or below percentile 5 and above percentile 5 for macular GCC thickness in healthy children aged 6–18 years (n = 157).

Only the inferior sector of GCIPL achieved a significant level of agreement (κ≥0.6) between adult and pediatric database classification.

## Discussion

### Peripapillary retinal nerve fibre layer thickness

Pediatric normative databases of pRNFL have been developed by several studies using different brands of OCT, as shown in Table 8. Our findings were comparable to those of Yao et al. [18] but thicker than others that used either Cirrus or Spectralis OCT. Pierro et al. [34] assessed RNFL using six different brands of SD-OCT and found that Cirrus and Spectralis OCTs showed the thinnest RNFL values in all measurements, while Topcon OCT was the highest. They considered the differences might be due to the different standard diameters of circle scans with different SD-OCTs. They hypothesized that measurements with a 3.4 mm diameter closer around the disc, as with Topcon 3D-OCT, may explain the greater thickness of the nerve compared to Cirrus OCT, which worked on a 3.46 mm diameter [34].

The human RNFL loses approximately 5000 axons per year from birth to death, approximately 2500 per year before age 50, and 7500 per year after 50 [37]. We did not find a significant age effect on RNFL thickness, consistent with the majority of studies on children [16–18,

| Superior Temporal | | | |
|---|---|---|---|
| Reference value Pediatric | | Reference value Adult | |
| Percentile | n | n | Percentile |
| ≤p1 | 1 | 9 | ≤p1 |
| >p1 to ≤p5 | 6 | | |
| >p5 | 2 | | |
| ≤p1 | | 15 | >p1 to ≤p5 |
| >p1 to ≤p5 | | | |
| >p5 | 15 | | |
| ≤p1 | | 134 | >p5 to ≤p95 |
| >p1 to ≤p5 | | | |
| >p5 | 134 | | |
| Specific agreement (%) between Peadiatric and adult for ≤p5 = 45.16% | | | |

| Superior | | | |
|---|---|---|---|
| Reference value Pediatric | | Reference value Adult | |
| Percentile | n | n | Percentile |
| ≤p1 | 1 | 6 | ≤p1 |
| >p1 to ≤p5 | 5 | | |
| >p5 | | | |
| ≤p1 | | 17 | >p1 to ≤p5 |
| >p1 to ≤p5 | | | |
| >p5 | 17 | | |
| ≤p1 | | 135 | >p5 to ≤p95 |
| >p1 to ≤p5 | | | |
| >p5 | 135 | | |
| Specific agreement (%) between Peadiatric and adult for ≤p5 = 41.38% | | | |

| Superior Nasal | | | |
|---|---|---|---|
| Reference value Pediatric | | Reference value Adult | |
| Percentile | n | n | Percentile |
| ≤p1 | 1 | 7 | ≤p1 |
| >p1 to ≤p5 | 5 | | |
| >p5 | 1 | | |
| ≤p1 | | 23 | >p1 to ≤p5 |
| >p1 to ≤p5 | 2 | | |
| >p5 | 21 | | |
| ≤p1 | | 128 | >p5 to ≤p95 |
| >p1 to ≤p5 | | | |
| >p5 | 128 | | |
| Specific agreement (%) between Peadiatric and adult for ≤p5 = 42.11% | | | |

| Inferior Nasal | | | |
|---|---|---|---|
| Reference value Pediatric | | Reference value Adult | |
| Percentile | n | n | Percentile |
| ≤p1 | 1 | 5 | ≤p1 |
| >p1 to ≤p5 | 4 | | |
| >p5 | | | |
| ≤p1 | | 11 | >p1 to ≤p5 |
| >p1 to ≤p5 | 1 | | |
| >p5 | 10 | | |
| ≤p1 | | 142 | >p5 to ≤p95 |
| >p1 to ≤p5 | | | |
| >p5 | 142 | | |
| Specific agreement (%) between Peadiatric and adult for ≤p5 = 54.55% | | | |

| Inferior | | | |
|---|---|---|---|
| Reference value Pediatric | | Reference value Adult | |
| Percentile | n | n | Percentile |
| ≤p1 | 1 | 2 | ≤p1 |
| >p1 to ≤p5 | 1 | | |
| >p5 | | | |
| ≤p1 | | 13 | >p1 to ≤p5 |
| >p1 to ≤p5 | 6 | | |
| >p5 | 7 | | |
| ≤p1 | | 143 | >p5 to ≤p95 |
| >p1 to ≤p5 | | | |
| >p5 | 143 | | |
| Specific agreement (%) between Peadiatric and adult for ≤p5 = 69.57% | | | |

| Inferior Temporal | | | |
|---|---|---|---|
| Reference value Pediatric | | Reference value Adult | |
| Percentile | n | n | Percentile |
| ≤p1 | 1 | 7 | ≤p1 |
| >p1 to ≤p5 | 5 | | |
| >p5 | 1 | | |
| ≤p1 | | 17 | >p1 to ≤p5 |
| >p1 to ≤p5 | | | |
| >p5 | 17 | | |
| ≤p1 | | 134 | >p5 to ≤p95 |
| >p1 to ≤p5 | | | |
| >p5 | 134 | | |
| Specific agreement (%) between Peadiatric and adult for ≤p5 = 40% | | | |

**Fig 4. Agreement between adult and pediatric OCT reference value classification.** Specific agreement between adult and pediatric percentiles equal to or below percentile 5 and above percentile 5 for macular GCIPL thickness in healthy children aged 6–18 years (n = 157).

35, 38–41]. This suggests that RNFL thickness remains stable in early childhood and adolescence [42]. Another possible explanation is that thinning of RNFL with age happens later in life. In the Taiwanese Chinese population, the earliest age-related RNFL thinning was found after 35 years, and age-related RNFL thinning was more significant in participants older than 41 years [43]. Thus, one may postulate that the age-related loss of RNFL thickness may not be linear but that the age-related RNFL thinning starts in adulthood.

Several studies reported a significant positive association between RNFL thickness and SE, though few studies showed no significant association between RNFL thickness and SE [16, 21, 38]. Our results agreed with other studies that reported thinner RNFL with higher myopic refraction [2, 35, 39, 41, 44, 45]. Barrio-Barrio et al. [35] showed a 1.05 μm change in RNFL thickness per diopter (D) change in SE; Rao et al. [44] reported a 1.9 μm reduction in RNFL full circle for every 1D of myopic shift; Ayala & Ntoula [45] stated a 2.1 μm increase in mean total RNFL thickness for every diopter towards hyperopia; Gürağaç et al. [2] reported the highest change with a 2.9 μm reduction in average RNFL thickness for every 1D myopic shift. Our results showed a 1.37 μm change in total RNFL thickness for every diopter change in SE. For sectorial RNFL changes, the most significant association of change was in inferior RNFL with 2.27 μm, followed by nasal RNFL with 2.20 μm, with every diopter change in SE. Thinner RNFL with more myopic refractive error may be explained by mechanical stretching and thinning of the retina due to elongation of the globe [46]. Another possibility is that OCT

**Table 8. Comparison of pediatric RNFL thickness reported in the literature with the current study.**

| Source | OCT | Population | Mean age ± SD (age range) | Mean RNFL thickness (µm) ± SD | | | | |
|---|---|---|---|---|---|---|---|---|
| | | | | Average | Superior | Nasal | Inferior | Temporal |
| Barrio-Barrio et al. 2013 [35] | Cirrus | Spanish | 9.58 ± 3.12 (4–17) | 97.40 ± 9.00 | 124.70 | 69.70 | 128.00 | 67.40 |
| Gürağaç et al. 2017 [2] | Cirrus | Turkish | 10.2 ± 4.10 (3–17) | 96.49 ± 10.10 | 122.29 ± 16.88 | 70.03 ± 10.78 | 125.82 ± 17.76 | 67.60 ± 9.93 |
| Goh et al. 2017 [21] | Cirrus | Singapore | 9.47 ± 3.40 (3–18) | 99.00 ± 11.45 | 123.20 ± 25.81 | 72.86 ± 1.14 | 124.24 ± 22.23 | 75.80 ± 1.37 |
| Raffa & AlSwealh 2023 [36] | Cirrus | Saudi | 10.19 ± 4.00 (4–18) | 93.88 ± 10.00 | 119.51 ± 14.50 | 69.99 ± 10.90 | 120.25 ± 16.90 | 67.07 ± 11.10 |
| Söhnel et al. 2023 [16] | Spectralis | German | 12.91 ± 3.29 (4 - <19) | 102.88 ± 8.79 | Superior temporal:145.80 Superior nasal: 108.43 | 74.50 | Inferior nasal: 107.53 Inferior temporal: 150.04 | 80.73 |
| Nemeş-Drăgan et al. 2023 [20] | Spectralis | Romania | 10.21 ± 8.32 (4–18) | 104.03 ± 11.42 | 132.73 ± 19.10 | 74.20 ± 16.48 | 134.64 ± 21.77 | 74.67 ± 11.95 |
| | Revo 80 | | 9.11 ± 7.20 (4–18) | 127.05 ± 15.61 | 144.44 ± 19.25 | 96.49 ± 19.41 | 144.86 ± 23.12 | 77.00 ± 11.40 |
| Yao et al. 2021 [18] | Topcon 3D OCT-1 | China (Tibetan) | 6.83 ± 0.46 | 112.86 ± 11.94 | 144.90 ± 17.60 | 79.51 ± 16.00 | 146.99 ± 19.13 | 79.99 ± 12.88 |
| Current study | Topcon 3D OCT-1 | Malaysia | 11.64 ± 3.31 (6–18) | 112.05 ± 8.65 | 145.11 ± 14.51 | 79.13 ± 12.78 | 144.89 ± 15.61 | 79.32 ± 11.06 |

measurements are affected optically by greater axial length and higher myopia, producing thinner RNFL as an artefact [47].

A substantial agreement between adult and pediatric database classification was observed in the temporal (65%) sectors of RNFL, and a moderate agreement was observed in the superior (54%), nasal (49%), and inferior (51%) RNFL. According to Ranganathan et al. (2017) [32], a significant level of agreement is achieved with a minimum of 60%. Hence, it appears that the current Topcon 3D-OCT-1 Maestro 2 database would be most suitable for sector-specific RNFL (temporal) measurement in children.

After comparing our results with previous studies performed with the same Topcon OCT device in children [18] and adults [3, 48], we found similar RNFL thickness results to those of Yao and colleagues [18]. RNFL thickness was slightly thicker in children compared to adults, except for nasal RNFL. In their investigations in children and adults, Yao et al. [18] and Chalagsian et al. [3, 48] found, respectively, the following values: total RNFL, 112.86 ± 11.94 µm and 104.04 ± 11.34 µm; superior RNFL, 144.90 ± 17.60 µm and 126.7 ± 17.77 µm; nasal RNFL, 79.51 ± 16.00 µm and 79.50 ± 16.46 µm; inferior RNFL, 146.99 ± 19.13 µm and 136.56 ± 17.42 µm; temporal RNFL, 79.99 ± 12.88 µm and 73.53 ± 11.81 µm. As RNFL thickness tends to be thicker in children than in adults, it causes the overdiagnosis of normality in children when the adult database is used for classification. Our results showed that 36 eyes (54 measurements) were considered abnormal based on the pediatric database, and 23 eyes (34 measurements) were classified as the same color category with the reference adult database (64%). Therefore, the likelihood of missing abnormal measurements in children when using the current OCT software was 36%, as extreme measurements can be overlooked.

## Macular thickness

In this study, the average macular thickness was 280.24 ± 12.46 µm. Other studies using Cirrus and Topcon OCTs reported an average macular thickness of 275–284 µm, which closely

**Table 9. A comparison of pediatric average macular thickness and macular volume in the literature with the current study.**

| Source | OCT | Population | Mean age ± SD (age range) | Average macula thickness (µm) ± SD | Macular volume mm³ ± SD[a] |
|---|---|---|---|---|---|
| Barrio-Barrio et al. 2013 [35] | Cirrus | Spanish | 9.58 ± 3.12 (4–17) | 283.62 ± 14.08 | 10.22 ± 0.49 |
| Al-Haddad et al. 2013 [49] | Cirrus | White Middle Eastern | 10.7 ± 3.14 (6–17) | 279.6 ± 12.5 | 10.1 ± 0.5 |
| Gürağaç et al. 2017 [2] | Cirrus | Turkish | 10.2 ± 4.10 (3–17) | 279.27 ± 12.59 | 9.97 ± 0.44 |
| Raffa & AlSwealh 2023 [36] | Cirrus | Saudi | 10.19 ± 4.00 (4–18) | 275.9 ± 17.7 | 9.9 ± 0.6 |
| Jammal et al. 2022 [50] | Primus | Jordan | 10.8 ± 3.0 (6–16) | 277.2 ± 12.5 | 10.0 |
| Söhnel et al. 2023 [16] | Spectralis | German | 12.91 ± 3.29 (4 - <19) | 320.53 ± 12.29 | 8.88 ± 0.35 |
| Muñoz-Gallego et al. 2021 [29] | Topcon 3D OCT-2000 | Caucasian | 10.26 ± 3.37 (5–18) | 275.42 ± 14.79 | 7.80 ± 0.35 |
| Yao et al. 2021 [18] | Topcon 3D OCT-1 | China (Tibetan) | 6.83 ± 0.46 | 279.19 ± 10.61 | 7.89 ± 0.30 |
| Current study | Topcon 3D OCT-1 | Malaysia | 11.77 ± 3.22 (6–18) | 280.24 ± 12.46 | 7.92 ± 0.35 |

matched the value we reported, except the study by Söhnel et al. [16], who used Spectralis OCT, which reported a higher value of 320.53 µm of average macular thickness. The total macular volume in our study was 7.92 ± 0.35 mm³, similar to other studies using Topcon OCT but lower than those obtained by Spectralis and Cirrus (9–10 mm³). Table 9 shows the average macular thickness and volume in the pediatric population in our study compared to other studies reported in the literature.

In our study, age was not correlated significantly with any macular parameters. Some studies demonstrated macular thickness positively correlated with age [16, 17, 24, 35, 40, 49, 51], while others did not [29, 52]. A few studies reported no significant correlation between macular thickness and SE [16, 38]. However, we found SE was positively correlated with average macular thickness, macular total volume, and all macular sectors in the ETDRS grid except centre ETDRS and minimum foveal thickness. Similar findings have previously been reported [2, 29, 49, 50]. We observed the perifoveal macular (outer ring) appeared to be more impacted by changes in refractive error compared to the parafoveal (inner ring) macular, while the central fovea was unaffected. Thinning of the peripheral retina may serve to compensate for the stretching force experienced across the entire retina caused by axial length elongation (more myopic in SE refraction), thereby preserving the optimal visual function of the central macular [53].

Comparing our macular thickness values with findings in adults [3, 48], our values of the centre and parafoveal macular (inner ETDRS) are slightly thinner or similar to those from Chalagsian et al. [3], which suggests that the development of center foveal and parafoveal is still ongoing in our pediatric subjects. However, our results showed a thicker perifoveal macular (outer ETDRS) than theirs. One might assume that the perifoveal macular in our subjects might be approaching/reaching adults' thickness, but the thinning due to aging has yet to start. Hence, the agreement between adult and pediatric database classification was found to be better in the perifovea macular (outer ETDRS) than parafoveal macular (inner ETDRS) and centre ETDRS.

The perifoveal macular (outer ring) revealed moderate to substantial agreement between adult and pediatric database classification: moderate agreement for outer nasal (49%) and

outer temporal (44%) quadrants and substantial agreement for outer superior (67%) and outer inferior (63%) quadrants. However, a slight or fair agreement was found in the parafoveal macular (inner ring) and centre macular between adult and pediatric database classification. The centre (16%) had a slight agreement while inner superior (24%), inner nasal (23%), inner inferior (21%), and inner temporal (33%) quadrants had a fair agreement. As for the specific agreement between pediatric and adult for $\leq$ p5 and $>$ p95, generally, 50% or less of the agreement was noticed for centre ETDRS and all inner ETDRS quadrants, but a higher percentage of agreement was shown for outer ETDRS.

From the total discrepancies of 289 measurements, 97.23% (281 measurements) were assigned to the higher percentile category with the pediatric database, while the remaining 2.77% (8 measurements) were assigned to the lower percentile category. A total of 134 measurements were considered abnormal ($\leq$p5 and $>$p95) based on the pediatric database, and only 41 measurements (30.60%) were classified as the same color category with the reference adult database. Consequently, the current OCT software, which overestimates macular thinness in pediatrics, may lead to discrepancies where children's macular appears to have normal thickness according to a pediatric database but is categorized as thin when compared to the adult database. This could lead to misdiagnosis such as macular thinning, unnecessary or inappropriate treatment for the actual condition, skew the monitoring of disease progression or response to treatment, unnecessary follow-up appointments or performing additional diagnostic tests, or cause patient anxiety. Therefore, an age-matched database is crucial for the pediatric population in OCT measurements.

**GCC and GCIPL thickness.** Unlike peripapillary RNFL and macula thickness, literature on developing normative databases for GCC and GCIPL thickness in normal healthy children using SD-OCT is scarce. For GCC thickness, our study showed a higher value of superior and inferior GCC thickness compared to those obtained by iVue-100 OCT, as shown in Table 10.

**Table 10. Comparison of pediatric GCC and GCIPL thickness reported in the literature with the current study.**

| Source | OCT | Population | Mean age ± SD (age range) | Mean thickness (μm) ± SD | | | | | |
|---|---|---|---|---|---|---|---|---|---|
| | | | | Superior temporal | Superior | Superior nasal | Inferior nasal | Inferior | Inferior temporal |
| **GCC** | | | | | | | | | |
| Current study | Topcon 3D OCT-1 | Malaysia | 11.65 ±3.27 (6–18) | 96.13 ± 6.26 | 109.46 ± 6.96 | 120.17 ± 7.43 | 120.85 ± 7.98 | 107.91 ± 6.92 | 98.78 ± 5.98 |
| Yao et al. 2021 [18] | Topcon 3D OCT-1 | China (Tibetan) | 6.83 ± 0.46 | 98.03 ± 7.39 | 109.58 ± 6.84 | 116.87 ± 6.95 | 116.01 ± 7.10 | 107.29 ± 6.88 | 101.03 ± 7.59 |
| Wang et al. 2021 [54] | iVue-100 | China | 7.10 ± 0.41 | NA[a] | 95.36 ± 8.07 | NA[a] | NA[a] | 95.27 ± 7.96 | NA[a] |
| **GCIPL** | | | | | | | | | |
| Current study | Topcon 3D OCT-1 | Malaysia | 11.65 ±3.27 (6–18) | 72.77 ± 5.57 | 72.77 ± 5.02 | 77.83 ± 5.71 | 76.47 ± 5.36 | 69.51 ± 4.70 | 74.30 ± 5.42 |
| Yao et al. 2021 [18] | Topcon 3D OCT-1 | China (Tibetan) | 6.83 ± 0.46 | 74.99 ± 6.18 | 74.43 ± 4.85 | 80.89 ± 5.70 | 80.44 ± 5.68 | 71.15 ± 4.58 | 76.52 ± 6.28 |
| Arnljots et al. 2020 [55] | Cirrus | Swedish | 6.5 | 84.5 ± 5.6 | 86.1 ± 6.2 | 87.1 ± 5.8 | 86.1 ± 5.5 | 85.1 ± 5.5 | 86.3 ± 5.4 |
| Galdos et al. 2014 [56] | Cirrus | Spanish | 9.6 ± 3.13 (4–17) | 83.9 | 85.9 | 86.5 | 84.7 | 83.0 | 84.3 |
| Muñoz-Gallego et al. 2019 [30] | Topcon 3D OCT-2000 | Caucasian | 10.3 ± 3.4 (5–18) | NA[a] | 73.70 ± 5.36 | NA[a] | NA[a] | 73.32 ± 4.76 | NA[a] |
| Goh et al. 2017 [21] | Cirrus | Singapore | 9.47 ± 3.40 (3–18) | NA[a] | 83.68 ± 6.96 | NA[a] | NA[a] | 81.64 ± 6.70 | NA[a] |

[a]NA, not available.

By contrast, our study reported approximate similar GCIPL thickness with other studies using Topcon OCT but much lower than those obtained by Cirrus OCT.

There was no significant correlation between age and GCIPL thickness, consistent with previously reported results [21, 24, 30], but a positive correlation between age and nasal GCC thickness (superior nasal and inferior nasal) was observed in our study. We found a negative correlation between GCC thickness and hyperopic SE and a positive correlation between GCIPL thickness and myopic SE. When the correlation was analyzed as a whole, our study showed GCC (superior temporal and inferior temporal) and GCIPL thickness had a positive correlation with SE, consistent with the findings of previous studies [18, 22, 57, 58], Cheng et al. [22] noted SE was the factor most closely related to GCIPL thickness compared to other variables such as age, gender, and axial length. Similarly, SE was found to be the only factor associated with GCIPL thickness in our study. From the model by Cheng et al. [22], every diopter increase in SE was associated with a 0.61 μm increase in GCIPL thickness and a 0.15 μm increase in GCC thickness. Yao et al. [58] recently reported myopic students had thinner GCIPL and GCC thickness in the outer ETDRS (perifoveal), and both GCIPL and GCC were correlated with SE. They suggested that the thickness of the outer ETDRS GCC and GCIPL thickness were more sensitive to SE changes and could be seen as potential risk factors for myopia progression.

Comparing our study results with those of previous studies performed with the same Topcon OCT device in children [18] and adults [3, 48], we found GCC and GCIPL thicknesses to be slightly thicker in children compared to adults, where one may expect the overdiagnosis of normality in children when using the adult database. Surprisingly, the opposite results were noted in this study. The detailed breakdown of the classification of adult and pediatric databases into three different percentile-based categories (Figs 3 & 4) showed that all the discrepancies in classification were assigned to lower percentile categories with adult databases for both GCC (120 measurements) and GCIPL (113 measurements).

The inferior hemisphere of GCC and GCIPL (except inferior temporal GCIPL) revealed better agreement between adult and pediatric database classification compared to the superior hemisphere. For GCC, a moderate agreement was observed in the inferior (58%), inferior nasal (52%), superior (48%), and inferior temporal (47%) quadrants, and fair agreement in the superior temporal (38%) and superior nasal (32%) quadrant. For GCIPL, a substantial agreement was observed in the inferior (67%) GCIPL quadrant; a moderate agreement in the inferior nasal (45%) GCIPL quadrants; a fair agreement in the superior temporal (39%), superior (38%), superior nasal (34%), and inferior temporal (28%) GCIPL quadrants. As for the specific agreement between pediatric and adult for ≤p5, the inferior hemisphere of GCC, superior GCC and inferior GCIPL quadrants achieved more than 50% of the agreement, while superior temporal GCC, superior nasal GCC quadrants and all GCIPL quadrants (except inferior quadrant) achieved less than 50% of the agreement.

The current OCT software, which tends to overestimate the thinness of GCC and GCIPL in children, can result in discrepancies. For instance, a child's GCC and GCIPL may appear to have normal thickness according to a pediatric database but might be categorized as thin when compared to the adult database. This could lead to misdiagnosis of conditions such as glaucoma or optic neuropathies, unnecessary diagnostic tests or inappropriate treatment for the actual condition, or cause patient anxiety. Hence, having an age-matched database specifically tailored for the pediatric population is essential for accurate OCT measurements.

There are some limitations in our study. Due to the study's cross-sectional design, we cannot ascertain the changes over time in OCT measurements in a specific child's eye. In addition, we did not measure axial length, which limited our ability to identify the relationship between RNFL and macular thickness with axial length. Hence, a longitudinal study including axial

length measurement on an equal number of different ethnic subjects is needed to confirm the effect of age, ethnicity, and axial length on retinal thickness in multiethnic children.

## Conclusion

The overall agreement for pRNFL and macular thickness measurements in children with the adult reference database was between 72% and 90%. Children's retinal thickness was not significantly correlated with age but positively associated with spherical equivalent. When using the inbuilt adult database in the SD-OCT as a reference, there is a tendency to miss abnormal RNFL measurements but overestimate the thinness of the macular, GCC, and GCIPL in children. Thus, when evaluating ophthalmic disorders in children using SD-OCT, it is advisable to utilize a pediatric reference database, such as the one established in this research for Topcon 3D OCT-1 Maestro 2, to improve the precision and efficiency of diagnosing and treatment plans customized to meet the individual requirements of pediatric patients.

## Author Contributions

**Conceptualization:** Mohd Izzuddin Hairol.

**Data curation:** Tian Siew Pua.

**Formal analysis:** Tian Siew Pua.

**Funding acquisition:** Mohd Izzuddin Hairol.

**Investigation:** Mohd Izzuddin Hairol.

**Methodology:** Tian Siew Pua, Mohd Izzuddin Hairol.

**Project administration:** Tian Siew Pua, Mohd Izzuddin Hairol.

**Resources:** Mohd Izzuddin Hairol.

**Software:** Mohd Izzuddin Hairol.

**Supervision:** Mohd Izzuddin Hairol.

**Validation:** Tian Siew Pua, Mohd Izzuddin Hairol.

**Visualization:** Mohd Izzuddin Hairol.

**Writing – original draft:** Tian Siew Pua.

**Writing – review & editing:** Mohd Izzuddin Hairol.

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
