## [Decision Letter · Decision Letter 0]

27 Aug 2024

PONE-D-24-23425Optical coherence tomography measurements in children: Exploring agreement with an adult reference database and the association spherical equivalent and age on retinal thicknessPLOS ONE

Dear Dr. Hairol,

Thank you for submitting your manuscript to PLOS ONE. After careful consideration, we feel that it has merit but does not fully meet PLOS ONE’s publication criteria as it currently stands. Therefore, we invite you to submit a revised version of the manuscript that addresses the points raised during the review process.

Title needs to be reframed.

Abstract has plenty of sentence formation errors which necessitates it to be rewritten.

“There is a tendency to overestimate the thinness of the macular, GCC, and GCIPL in children when using databases currently included in the SD-OCT.” This sentence in the Conclusion section needs to be reframed.

We look forward to receiving your revised manuscript.

Kind regards,

Kumar Saurabh

Academic Editor

PLOS ONE

2. In the online submission form you indicate that your data is not available for proprietary reasons and have provided a contact point for accessing this data. Please note that your current contact point is a co-author on this manuscript. According to our Data Policy, the contact point must not be an author on the manuscript and must be an institutional contact, ideally not an individual. Please revise your data statement to a non-author institutional point of contact, such as a data access or ethics committee, and send this to us via return email. Please also include contact information for the third party organization, and please include the full citation of where the data can be found.

Additional Editor Comments:

Title needs to be reframed.

Abstract has plenty of sentence formation errors which necessitates it to be rewritten.

“There is a tendency to overestimate the thinness of the macular, GCC, and GCIPL in children when using databases currently included in the SD-OCT.” This sentence in the Conclusion section needs to be reframed.

Reviewers' comments:

Reviewer's Responses to Questions

**Comments to the Author**

1. Is the manuscript technically sound, and do the data support the conclusions?

Reviewer #1: Yes

Reviewer #2: Yes

2. Has the statistical analysis been performed appropriately and rigorously? 

Reviewer #1: Yes

Reviewer #2: Yes

3. Have the authors made all data underlying the findings in their manuscript fully available?

Reviewer #1: Yes

Reviewer #2: Yes

4. Is the manuscript presented in an intelligible fashion and written in standard English?

Reviewer #1: Yes

Reviewer #2: Yes

5. Review Comments to the Author

Reviewer #1: Congratulations on a well-written article. I have a few comments.

-it would be useful if the macular thickness association with the spherical equivalent was presented in tabular form for ease of understanding and better corelation with degree of refractive error. the data needs better presentation.

-The topic of RNFL and macular thickness in pediatric population has been well researched, your study is well done but it would be even more useful if these measurements were analysed for specific population and a detailed correlation with spherical equivalent was provided

Reviewer #2: Thank you for allowing me to read your study. The authors have evaluated the peripapillary nerve fibre layer and macular thickness in 160 paediatric eyes and the correlation between retinal thickness and age and spherical equivalent. This is an exhaustive study where the authors have considered factors such as birth weight and taken only children with birth weight >2500 g. Axial length has not been included in this study and has been mentioned in the limitations.

The overall agreement between the findings in paediatric population and the available adult reference database was 73-88%. For macular thickness it was >73%, for peripapillary RNFL thickness, it was >88% and for GCC and GCIPL, it was 84-86%. A change of 1.43micron in RNFL thickness was noted for every 1D change of SE. GCC and GCIPL was found to be thicker in children leading to possible overdiagnosis of normality in children with the adult database.

While there have been similar studies conducted in different populations, like Chinese, Tibetan, German, Romanian etc, this aims to give a reference database in the Malaysian paediatric population. If possible, the authors can briefly summarise the average findings of the various studies in different populations in the form of a table.

6. PLOS authors have the option to publish the peer review history of their article (what does this mean?). If published, this will include your full peer review and any attached files.

Reviewer #1: No

Reviewer #2: No

---

## [Author Response · Author response to Decision Letter 0]

13 Sep 2024

Response to Editor

1. Title needs to be reframed.

Our response: the title has been reframed as

“Evaluating Retinal Thickness Classification in Children: A Comparison Between Pediatric and Adult Optical Coherence Tomography Databases”

 2. Abstract has plenty of sentence formation errors which necessitates it to be rewritten.

Our response: The Abstract has been rewritten.

 3. “There is a tendency to overestimate the thinness of the macular, GCC, and GCIPL in children when using databases currently included in the SD-OCT.” This sentence in the Conclusion section needs to be reframed.

Our response: the sentence has been reframed as

“When using the inbuilt adult database in the SD-OCT as a reference, there is a tendency to miss abnormal RNFL measurements but overestimate the thinness of the macular, GCC, and GCIPL in children.”

4. Please review your reference list to ensure that it is complete and correct.

Our response: Reference list has been corrected accordingly.

Response to Reviewer #1

1. It would be useful if the macular thickness association with the spherical

 equivalent was presented in tabular form for ease of understanding and better correlation with degree of refractive error. the data needs better presentation.

Our response: The table has been added at

Line 238: Table 4. Correlation between macular parameters with age and SE

2. The topic of RNFL and macular thickness in pediatric population has been well researched, your study is well done but it would be even more useful if these measurements were analysed for specific population and a detailed correlation with spherical equivalent was provided

Our response: Correlation with spherical equivalent by SE groups has been added.

For RNFL:

 Line 192-193: Correlations between SE groups and RNFL thickness are shown in Table 2, where a significant correlation was found only in the myopic SE group.

Line 194: Table 2. Correlation between RNFL thickness with SE groups

For Macular:

Line 243-245: Statistically significant positive correlations were found in myopic SE for all the macular parameters except minimum foveal and centre ETDRS thickness. No significant correlation was found between the emmetrope and hyperope groups, as shown in Table 5.

Line 246: Table 5. Correlation between macular parameters with SE groups

For GCC & GCIPL:

Line 290-295: A significant weak positive correlation was found between inferior and temporal GCC and myopic SE; but a significant moderate negative correlation was found between all sectors of GCC and hyperopic SE. For GCIPL, significant weak to moderate positive correlations were found with myopic SE for all sectors. The summary of the correlations between GCC and GCIPL with refraction groups is shown in Table 7.

Line 296: Table 7. Correlation between GCC and GCIPL with SE groups

Discussion: Lines 466-470

Response to Riewer #2

While there have been similar studies conducted in different populations, like Chinese, Tibetan, German, Romanian etc, this aims to give a reference database in the Malaysian paediatric population. If possible, the authors can briefly summarise the average findings of the various studies in different populations in the form of a table.

Our response: The average findings of various studies have been added to the tables and discussed in Discussion.

RNFL: Discussion at Lines 341-349

Lines 350-351: Table 8. Comparison of pediatric RNFL thickness reported in the literature with the current study.

Macular: Discussion at Lines 398-405

Lines 406-407: Table 9. Comparison of pediatric average macular thickness and macular volume reported in the literature with the current study.

GCC & GCIPL: Discussion at Lines 454-459

Line 460-461: Table 10. Comparison of pediatric GCC and GCIPL thickness reported in the literature with the current study.

---

## [Decision Letter · Decision Letter 1]

11 Nov 2024

Evaluating Retinal Thickness Classification in Children: A Comparison Between Pediatric and Adult Optical Coherence Tomography Databases

PONE-D-24-23425R1

Dear Dr. Hairol,

We’re pleased to inform you that your manuscript has been judged scientifically suitable for publication and will be formally accepted for publication once it meets all outstanding technical requirements.

Kind regards,

Kumar Saurabh

Academic Editor

PLOS ONE

Additional Editor Comments (optional):

Reviewers' comments:

Reviewer's Responses to Questions

**Comments to the Author**

1. If the authors have adequately addressed your comments raised in a previous round of review and you feel that this manuscript is now acceptable for publication, you may indicate that here to bypass the “Comments to the Author” section, enter your conflict of interest statement in the “Confidential to Editor” section, and submit your "Accept" recommendation.

Reviewer #2: All comments have been addressed

Reviewer #3: All comments have been addressed

2. Is the manuscript technically sound, and do the data support the conclusions?

Reviewer #2: Yes

Reviewer #3: Yes

3. Has the statistical analysis been performed appropriately and rigorously? 

Reviewer #2: Yes

Reviewer #3: Yes

4. Have the authors made all data underlying the findings in their manuscript fully available?

Reviewer #2: Yes

Reviewer #3: Yes

5. Is the manuscript presented in an intelligible fashion and written in standard English?

Reviewer #2: Yes

Reviewer #3: (No Response)

6. Review Comments to the Author

Reviewer #2: (No Response)

Reviewer #3: The authors appeared to have responded clearly to all the comments made by previous reviewers. I have no additional questions or issues. With the modifications in the way the results have been presented (read addition of tables) and changes made in Discussion and references, based on suggestions of previous reviewers, I found no additional issues

7. PLOS authors have the option to publish the peer review history of their article (what does this mean?). If published, this will include your full peer review and any attached files.

Reviewer #2: No

Reviewer #3: No

---

## [Editor Report · Acceptance letter]

12 Dec 2024

PONE-D-24-23425R1 

PLOS ONE

Dear Dr. Hairol, 

I'm pleased to inform you that your manuscript has been deemed suitable for publication in PLOS ONE. Congratulations! Your manuscript is now being handed over to our production team.

Kind regards, 

on behalf of

Dr. Kumar Saurabh 

Academic Editor

PLOS ONE